# Significance of EVs in Prostate Cancer Bone Metastases

**DOI:** 10.3390/ijms262412160

**Published:** 2025-12-18

**Authors:** Kagenori Ito, Takaaki Tamura, Fumihiko Urabe, Shinichi Sakamoto, Takahiro Kimura, Shin Egawa, Takahiro Ochiya

**Affiliations:** 1Department of Urology, The Jikei University School of Medicine, Tokyo 105-8471, Japan; ito.kagenori@gmail.com (K.I.); furabe0809@gmail.com (F.U.); tkimura0809@gmail.com (T.K.); egpro1981@gmail.com (S.E.); 2Department of Urology, Chiba University, Chiba 263-8522, Japan; turukameland@gmail.com (T.T.); rbatbat1@gmail.com (S.S.); 3Department of Molecular and Cellular Medicine, Institute of Medical Science, Tokyo Medical University, Tokyo 160-0023, Japan

**Keywords:** prostate cancer, bone metastasis, extracellular vesicles, osteoblast, osteoclast

## Abstract

Prostate cancer (PCa) exhibits a unique propensity to metastasize to bone, where it predominantly generates osteoblastic lesions. The formation of these lesions is a complex and dynamic process driven by reciprocal interactions between tumor cells and the bone microenvironment. Emerging evidence indicates that extracellular vesicles (EVs) play pivotal roles in the establishment of metastatic colonies and disease progression, as well as in local tumor–bone interactions. Through their diverse cargos, including proteins, lipids, and non-coding RNAs, EVs mediate bidirectional communication that regulates osteoclastogenesis, osteoblast activation, and osteocyte function, ultimately reshaping the bone niche to favor tumor growth. Importantly, EVs exhibit dual and context-dependent functions, acting either as promoters or suppressors of malignancy depending on the cellular source and microenvironmental context. These insights highlight EVs not only as mechanistic drivers of PCa bone metastases but also as promising therapeutic targets. Approaches aimed at modulating EV biogenesis, eliminating deleterious EVs, or harnessing EVs as drug delivery vehicles hold significant potential for advancing treatment strategies against PCa bone metastases.

## 1. Introduction

Extracellular vesicles (EVs) are nanoscale, membrane-bound particles secreted by virtually all cell types into various biological fluids, including blood, urine, saliva, breast milk, and bronchoalveolar lavage fluid [1]. Their presence was first noted in the early 1980s, when vesicular structures released during the maturation of erythrocytes were observed and, in studies of sheep reticulocytes, later termed “exosomes” by Johnson and colleagues [2,3,4]. These vesicles were initially regarded as a means of cellular disposal, carrying unwanted material out of the cell. A pivotal shift in understanding came in 1996, when Raposo et al. demonstrated that EVs derived from a B-cell lymphoma line could promote immune responses by enhancing antigen-presenting cell (APC) maturation and stimulating T-cell activation [5]. Subsequent discoveries revealed that EVs encapsulate functional biomolecules, including messenger RNAs and microRNAs, capable of modulating recipient cell phenotypes [1,6,7,8,9].

Research over the past decades has firmly established EVs as active mediators of intercellular communication. Within the tumor microenvironment (TME), they participate in diverse processes such as tumor initiation and progression, angiogenesis, formation of premetastatic niches, modulation of immune responses, and transfer of drug resistance traits [9]. Notably, organ-specific metastatic patterns have been linked to distinct integrin repertoires on tumor-derived EVs, as described by Hoshino et al., suggesting a role in directing metastatic organotropism [10].

EVs are generally classified according to size and biogenesis into three major subtypes: exosomes (approximately 40–160 nm in diameter, formed within multivesicular bodies), microvesicles (100–1000 nm, shed directly from the plasma membrane), and apoptotic bodies (500–2000 nm, generated during the late stages of apoptosis) [1,8,9]. While exosomes and microvesicles arise during physiological cellular activity, apoptotic bodies are a hallmark of programmed cell death. In practice, distinguishing these subtypes in biological samples is challenging, and in line with recommendations from the International Society for Extracellular Vesicles, the umbrella term “EVs” is used throughout this document [11].

Their remarkable stability in biofluids has spurred interest in EVs as a source for minimally invasive “liquid biopsy.” Indeed, EV-associated non-coding RNAs, such as microRNAs and long non-coding RNAs, have been implicated in cancer progression, metastatic dissemination, and resistance to systemic therapies [12,13,14,15,16,17,18,19].

Prostate cancer (PCa) is the second leading cause of cancer mortality among men in the United States, and roughly one in eight men will develop the disease during their lifetime [20]. PCa displays marked clinical and biological heterogeneity across age and stage. Although localized tumors are frequently controlled with definitive local therapy such as surgery or radiotherapy, outcomes for metastatic disease remain guarded, with a 5-year survival of about 55% [21]. Androgen-deprivation therapy (ADT) is the systemic backbone for locally advanced and metastatic PCa, yet most patients ultimately progress to castration-resistant PCa (CRPC). Cytotoxic chemotherapy improves overall survival in both metastatic hormone-sensitive PCa (mHSPC) and CRPC [22,23], but secondary resistance commonly emerges. Over the past decade, regulatory approvals of next-generation androgen receptor pathway inhibitors (ARSIs)—including abiraterone acetate and enzalutamide—and of the PARP inhibitor olaparib have reshaped the therapeutic landscape and expanded options for men with advanced disease [24,25].

Bone is the most frequent site of metastasis in metastatic CRPC, occurring in approximately 80% of patients [26,27]. PCa exhibits a predilection for the axial skeleton, where metastases can lead to a spectrum of skeletal-related events (SREs), including pain, pathological fractures, spinal cord compression, and other complications, all of which significantly impair quality of life and survival [28,29,30]. Although bone-targeting agents such as bisphosphonates and denosumab can mitigate SRE risk, median survival following the diagnosis of bone metastasis remains limited, ranging from 12 to 53 months [31,32].

The prostate gland physiologically secretes a specialized population of extracellular vesicles known as prostasomes, originally identified in seminal plasma. Prostasomes are enriched in cholesterol- and sphingomyelin-rich membranes, antioxidant enzymes, immunomodulatory proteins, and signaling molecules that regulate sperm function and local immune modulation [33]. PCa cells also release prostasome-like EVs, which have been implicated in promoting tumor progression, invasion, and metastatic dissemination, thereby providing a biological link between normal prostate EV biology and cancer-associated EV signaling [34].

In this review, we summarize current knowledge on the role of EVs in the development and progression of bone metastases in PCa and discuss how emerging EV research may inform and enhance clinical strategies for managing this challenging complication.

## 2. Molecular and Cellular Insights into Prostate Cancer Bone Metastasis

PCa is a malignancy that frequently develops bone metastases, predominantly characterized by osteoblastic lesions. In contrast, other cancers such as lung and breast cancer, which also metastasize to bone with high frequency, typically exhibit osteolytic bone metastases [35]. Histopathological analyses, however, have revealed that osteoblastic lesions in PCa are, in fact, mixed lesions accompanied by osteolytic processes [36,37]. Yonou et al. demonstrated in a mouse model that osteoclastic bone resorption precedes tumor proliferation [38], suggesting that osteoclast activation plays a pivotal role in the early stages of bone metastasis establishment. Clinically, patients with bone metastases are known to have an increased frequency of subsequent metastases to other organs and to lymph nodes beyond the regional basin [39], highlighting the role of bone metastases as pivotal mediators of systemic progression [28,40,41]. Understanding the unique mechanisms underlying bone metastasis in PCa is therefore crucial for elucidating disease biology and holds the potential to drive therapeutic innovation.

Historically, several hypotheses have been proposed regarding the mechanisms of metastasis formation. In 1928, James Ewing proposed the hemodynamic theory, which explained that metastases preferentially occur in organs with abundant blood flow [42]. In contrast, in 1940, Batson highlighted the role of the valveless vertebral venous plexus as a potential retrograde pathway, enabling PCa cells to reach the spine directly from pelvic organs [43]. However, in PCa, where bone metastases develop with high frequency despite the relatively low blood flow compared to organs such as the lung and liver, neither the hemodynamic theory nor the venous plexus theory sufficiently explains the clinical pattern of spread. Consequently, Paget’s “seed and soil” hypothesis, first proposed in 1889, is now considered more plausible, as it emphasizes the reciprocal interactions between tumor cells and the bone microenvironment as the foundation of metastasis formation [44].

Building upon these historical hypotheses, recent research has established a unified model based on the sequential processes of epithelial–mesenchymal transition (EMT), hematogenous dissemination, homing to the bone microenvironment, and the establishment of the vicious cycle, which ultimately drives the formation of osteoblastic bone metastases characteristic of PCa (Figure 1) [28,35,45].

Cancer primarily spreads systemically via hematogenous or lymphatic routes, leading to metastasis in multiple organs [46,47]. In PCa, the predominant pathway to bone metastasis is hematogenous dissemination [45]. Under the stimulation of factors such as Transforming growth factor (TGF)-β, Wnt, Epidermal Growth Factor (EGF), and Insulin-like growth factor (IGF), cancer cells undergo EMT, characterized by the downregulation of epithelial markers such as E-cadherin and cytokeratin, and the upregulation of mesenchymal markers including vimentin, N-cadherin, Snail, and Twist, supported by in vitro data [48]. These changes have also been observed in circulating tumor cells (CTCs) and disseminated tumor cells (DTCs), where the loss of epithelial markers and the acquisition of mesenchymal markers represent a hallmark of hematogenous dissemination, supported by clinical observations [49]. The EMT process is strongly supported by the presence of tumor-associated macrophages (TAMs) and cancer-associated fibroblasts (CAFs). TAMs secrete factors such as TGF-β, EGF, Interleukin (IL)-6, and Tumor Necrosis Factor (TNF)-α, thereby promoting EMT and invasion of cancer cells, supported by in vitro and in vivo studies [50]. In parallel, CAFs produce Hepatocyte growth factor (HGF), Fibroblast growth factor (FGF), and C-X-C motif chemokine ligand (CXCL)12, which enhance tumor cell motility and invasiveness, while also secreting Matrix metalloproteinase (MMP)s that degrade the basement membrane and extracellular matrix, thereby facilitating vascular intravasation, supported by in vitro and preclinical in vivo studies [51]. EVs also act as potent enhancers of EMT in this process, inducing E-cadherin downregulation and upregulation of the transcription factors Zinc finger E-box binding homeobox (ZEB)1/2 through the regulation of microRNA (miR)-21 and the miR-200 family, supported by in vitro studies and preclinical in vivo models [52,53]. In addition, tumor-derived EVs containing miR-210 have been shown to enhance invasive potential through hypoxia-driven responses, supported by in vivo and preclinical in vivo studies [54].

Hematogenously disseminated PCa cells home to the red bone marrow through interactions mediated by the CXCL12–C-X-C motif chemokine receptor (CXCR)4 axis derived from bone marrow stromal cells, as well as integrins, supported by in vivo and in vivo studies [55,56]. The bone microenvironment, characterized by sluggish blood flow and sinusoidal vessels, provides a favorable milieu for tumor cell arrest and adhesion [57,58]. Once homed to the bone, tumor cells and their EVs actively contribute to the establishment of a pre-metastatic niche. The concept of the pre-metastatic niche was first introduced by Kaplan et al. in 2005 [59], showing that tumor-derived factors and EVs can precondition distant organs to become receptive “soil” for metastasis. Consistently, Hoshino et al. reported that integrin expression patterns on EV surfaces dictate organotropic metastasis [10], suggesting that EVs may similarly contribute to the establishment of a pre-metastatic niche within the bone microenvironment in PCa bone metastasis (Figure 2).

Tumor cells that reach the bone do not possess the intrinsic ability to directly degrade mineralized bone tissue [45]. To establish metastatic foci, they must first create space for tumor expansion by destroying mineralized bone. This role is fulfilled by osteoclasts. Tumor cells secrete factors such as Parathyroid hormone-related protein (PTHrP), IL-6, IL-11, and Prostaglandin (PG)E2, which induce Receptor activator of NF-κB ligand (RANKL) expression in osteoblasts and thereby promote osteoclast differentiation, supported by in vitro and in vivo models [60]. Activated osteoclasts then resorb mineralized bone matrix, releasing growth factors such as TGF-β and IGF-1 in the process, supported by in vitro and in vivo models [61,62]. These factors act on tumor cells to enhance proliferation and further bone destruction, accelerating the formation of metastatic lesions [63]. This “osteoclast-dependent space formation” is considered an essential step in the establishment of bone metastases, including those from PCa, and serves as a pivotal event initiating the vicious cycle in which bone destruction and tumor growth mutually reinforce each other [64]. Our group has reported that PCa-derived EVs promote osteoclast formation in the presence of RANKL, supported by in vitro studies [65], suggesting that EVs function as auxiliary factors in osteoclast differentiation. Similarly, PCa-derived EVs have been shown to contain miR-21 and miR-378, which promote the differentiation of osteoclast precursors and thereby enhance bone resorptive activity, supported by in vitro and in vivo models [66,67]. Osteoclasts themselves also release EVs, which act on tumor cells and surrounding osteoblasts to facilitate remodeling of the bone microenvironment [68]. Thus, in the pathophysiology of PCa bone metastasis, EVs serve not only as mediators of intercellular communication but also as direct and indirect enhancers of osteoclastogenesis, thereby contributing to the establishment of the vicious cycle of bone resorption and tumor progression (Figure 2).

In PCa bone metastases, osteoblastic lesions emerge as a consequence of aberrant osteoblast activation following an initial osteolytic phase. Several molecular pathways have been identified as drivers of osteoblastic lesion formation. First, the endothelin-1 (ET-1)–ET_A receptor axis is involved. In PCa, ET-1 is produced and directly stimulates osteoblast proliferation and new bone formation, thereby driving the development of osteoblastic bone metastases, supported by in vitro, in vivo, and clinical observations [69,70]. Second, the dynamics of the Wnt signaling pathway and its antagonist Dickkopf-1 (DKK-1) are critical. In PCa, activation of the Wnt pathway promotes osteoblast differentiation and bone formation. Normally, DKK-1 acts as an inhibitor of this pathway, but reduced expression of DKK-1 releases this inhibition, thereby further enhancing bone-forming activity, as demonstrated in preclinical models [71]. Third, Bone morphogenetic protein (BMP) signaling, particularly BMP-6, is implicated. BMP-6 is highly expressed in PCa and has been linked to osteoblastic metastases, strongly promoting osteoblast differentiation, supported by in vitro and in vivo models [72]. In addition, prostate-specific antigen (PSA) also influences bone metabolism. PSA reduces the bioactivity of PTHrP through proteolytic cleavage, thereby attenuating osteoclast activity and favoring a bone-forming environment. PSA has also been reported to induce osteoprotegerin (OPG) production by osteoblasts, suppressing osteoclast activity via modulation of the RANKL/OPG axis, supported by in vitro studies [73]. Tumor-derived EVs have attracted attention as critical modulators of osteoblastic lesion development. PCa-derived EVs have been shown to enhance osteoblast differentiation, alkaline phosphatase activity, and the expression of mineralization markers, with Phospholipase D2 (PLD2)-dependent EV production being required for these effects, supported by in vitro and in vivo models [74]. In addition, EV cargo such as miR-18a-5p has been shown to stimulate the β-catenin pathway, thereby enhancing Wnt-driven osteogenic signaling, supported by in vitro studies [75]. Conversely, other studies have demonstrated that PCa-derived EVs may carry inhibitory cargo that suppresses osteoblast differentiation, suggesting a dual, context-dependent role. Notably, EVs may suppress osteoblast activity in the pre-metastatic stage but promote differentiation following tumor cell colonization, thereby exerting plastic, stage-specific functions [76]. Collectively, EVs act as fine-tuning regulators of osteoblastic lesion formation and are deeply involved in the pathophysiology of PCa bone metastases (Figure 2).

The metastatic niche is not merely a site of colonization for cancer cells but can serve as a platform for further malignant transformation and dissemination to distant organs. The bone matrix is rich in EMT-inducing factors such as TGF-β, IGF-1, BMP, and HGF, which are abundantly released during bone resorption. These factors have been shown to re-induce EMT in tumor cells, endowing them with stem-like properties and enhanced invasive potential, thereby increasing their ability to form multi-organ and distant lymph node metastases, supported by in vitro and in vivo models [77,78]. On the other hand, the bone metastatic microenvironment does not always promote tumor progression but may also exert suppressive effects. Our research group reported that Serpin family A member 3 (SERPINA3) and Lipocalin2 (LCN2), which are upregulated in PCa and modulated by osteoblast-derived EVs, regulate PCa cell proliferation and invasion in a tumor-suppressive manner, supported by in vitro and in vivo models [79]. These findings suggest that the education of tumor cells within bone metastases is not unilaterally directed toward 74malignancy but rather exhibits a dual nature of both promotion and suppression depending on the surrounding microenvironment and EV cargo. Understanding this duality will be essential for the development of future therapeutic strategies (Figure 2).

## 3. The Role of Osteoclast in Bone Metastasis

Although PCa bone metastases are classically described as osteoblastic, histopathological and clinical studies have demonstrated that osteolytic activity is an essential initial event that precedes osteoblastic lesion formation [36,37,38]. Indeed, in studies evaluating the clinical utility of serum bone turnover markers, markers reflecting osteoclast activity, such as serum type I collagen cross-linked C-terminal telopeptide (ICTP), have been shown to be more effective in predicting metastatic bone volume and patient survival than osteoblastic markers like serum Alkaline phosphatase (ALP) or Bone Specific Alkaline Phosphatase (BAP) [80]. Based on this accumulating body of evidence, therapies targeting osteoclasts are now routinely administered in clinical practice alongside hormonal therapy or chemotherapy for the treatment of patients with PCa bone metastases.

In addition to classical evidence for osteoclast activation in PCa bone metastases, recent studies have highlighted EVs as important mediators of tumor–bone interactions. Osteoclast-derived EVs carrying miR-21 and miR-214-3p can be directly transferred to prostate cancer cells, where they promote EMT, migration, and therapy resistance through modulation of Phosphatase and Tensin Homolog (PTEN)/AKT serine/threonine kinase 1 (AKT) signaling, supported by in vitro studies [81,82] (Figure 2).

Osteoclasts also regulate osteocytes through both direct and indirect mechanisms. Inflammatory cytokines such as TNF-α, IL-1β, and IL-6 secreted by activated osteoclasts increase sclerostin and RANKL expression while reducing OPG, thereby promoting bone resorption, supported by in vitro and in vivo models [83,84]. In parallel, osteoclast-derived EVs enriched in miR-214-3p impair osteocyte remodeling functions, and miR-21 further shifts the RANKL/OPG balance toward enhanced osteoclastogenesis, supported by in vitro and in vivo models [85,86]. Beyond these direct actions, osteoclastic bone resorption indirectly affects osteocytes by releasing growth factors such as TGF-β and IGF-1 stored in the bone matrix, which modulate osteocyte metabolism and signaling, supported by in vitro and in vivo models [87,88] (Figure 2).

In terms of osteoblast regulation, osteoclast-derived EVs enriched in miR-214 suppress osteoblast activity by acting through the EphrinA2/EPH receptor A2 (EphA2) axis [89] and by inhibiting Activating Transcription Factor 4 (ATF4), supported by in vitro and in vivo models [90]. Moreover, miR-214 also enhances osteoclastogenesis via Phosphatidylinositol-3 kinase (PI3K)–Akt signaling through PTEN suppression, supported by in vitro and in vivo models [91]. These findings suggest that osteoclast-derived EVs containing miR-214 may exert dual functions that collectively enhance bone destruction. Conversely, vesicles released from maturing osteoclasts containing membrane-bound RANK can activate RANKL reverse signaling in osteoblasts, leading to Runt-related transcription factor 2 (Runx2) induction and stimulation of bone formation, supported by in vitro and in vivo models [92] (Figure 2).

Based on this growing body of evidence, our research group was the first in the world to investigate the role of osteoclast-derived EVs in the presence of PCa. We found that osteoclasts influenced by PCa cells acquire malignant characteristics, with enhanced expression of genes associated with the pro-inflammatory cytokine IL-1β. These “malignant” osteoclasts release EVs that are enriched with microRNAs that further activate bone-resorbing cells and, conversely, suppress the function of osteoblasts responsible for bone formation. In a mouse model of prostate cancer bone metastasis, administration of microRNAs specifically derived from these malignant osteoclast EVs resulted in accelerated tumor progression and abnormal bone destruction. These findings suggest that osteoclasts abnormally activated at the invasive front of metastatic lesions contribute to further bone destruction, facilitating tumor progression in PCa bone metastases [93]. However, the EVs examined in this study were primarily within the small EV fraction, including exosomes. Other types of vesicles, such as large EVs including microvesicles and apoptotic bodies released upon the cell death of short-lived osteoclasts, may serve distinct functions. In fact, osteoclast-derived apoptotic bodies have been reported to act as coupling factors during normal bone remodeling, promoting osteoblast activation and thereby contributing to bone metabolism regulation [94]. The differential roles of various osteoclast-derived EV subtypes within the bone metastatic microenvironment remain largely unexplored and warrant further investigation.

## 4. The Role of Osteoblast in Bone Metastasis

PCa is unique among solid tumors in that its bone metastases are characterized by the formation of osteoblastic lesions accompanied by marked osteoblast proliferation [28]. Tumor cell–derived factors such as endothelin-1, BMPs, and Wnt signaling molecules have been shown to directly stimulate and activate osteoblasts, thereby enhancing their proliferation, supported by in vitro and in vivo models [95]. Moreover, PCa-derived EVs have been reported to contain Wnt pathway–activating molecules and miRNAs (e.g., miR-141 and miR-375), which stimulate osteoblasts, suggesting that EVs may serve as molecular mediators of osteoblast proliferation, supported by in vitro and in vivo models [75,96]. The effects of osteoblasts on the bone metastatic microenvironment have also been reported, as summarized below (Figure 2).

Osteoblasts not only function as bone-forming cells but also contribute to the metastatic process by providing diverse signals to tumor cells. Shiozawa et al. demonstrated that osteoblasts secrete cytokines such as IL-6, CXCL12, and IGF-1, which promote the proliferation, invasion, and even acquisition of drug resistance in PCa cells, supported by in vitro and in vivo models [56]. Through these mechanisms, bone metastatic sites function as a “niche” for tumor progression. Our research group further revealed that when osteoblast-derived EVs are internalized by PCa cells, they induce the expression of SERPINA3 and LCN2, exerting tumor-suppressive effects while simultaneously promoting osteoblastic metastasis, supported by in vitro and in vivo models [79]. Conversely, Liu et al. reported that osteoblast-derived exosomal miR-140-3p activates the AKT/mechanistic Target of Rapamycin (mTOR) pathway in PCa cells, suppresses autophagy, and thereby enhances proliferation, invasion, and migration, supported by in vitro and in vivo models [97]. These findings collectively suggest that osteoblast-derived EVs possess a “dual role,” exerting both tumor-suppressive and tumor-promoting effects depending on the tumor microenvironment and disease stage (Figure 2).

Osteoblasts play a dual role in regulating osteoclast differentiation. They promote osteoclastogenesis by expressing RANKL, while simultaneously suppressing it by secreting OPG. This opposing activity highlights their function as key modulators of bone resorption, supported by in vitro and in vivo models [98]. In PCa bone metastases, stimulation from tumor cells skews this balance toward enhanced RANKL expression, resulting in osteoclast activation and accelerated bone resorption, supported by in vitro and in vivo models [63]. Furthermore, it has been reported that osteoblast-derived EVs directly influence osteoclast differentiation. Cappariello et al. demonstrated that osteoblast-derived EVs contain inhibitory factors such as OPG that suppress osteoclast formation, supported by in vitro and in vivo models [99]. Consistently, our research group demonstrated that osteoblast-derived EVs significantly inhibit the formation of TRAP-positive multinucleated osteoclasts from RAW264.7 cells in the presence of RANKL, indicating that these EVs regulate osteoclastogenesis by suppressing osteoclast activity through a novel mechanism, supported by in vitro and in vivo models [79]. Taken together, these findings indicate that osteoblasts, in addition to secreting soluble factors (RANKL/OPG), flexibly regulate osteoclast activation or suppression through EV-mediated signaling, depending on tumor-derived stimuli and the specific conditions of the bone microenvironment (Figure 2).

TGF-β and IGF-1 are stored in the bone matrix produced by osteoblasts and are released during osteoclastic bone resorption, thereby creating a vicious cycle that enhances tumor cell proliferation and survival, supported by in vitro and in vivo models [87,88]. In recent years, osteoblast-derived EVs have also been reported to directly participate in this remodeling process. Cappariello et al. demonstrated that osteoblast-derived EVs contain MMP-13 and ALP, which directly act on the bone matrix to regulate remodeling and mineralization processes, supported by in vitro studies [99]. Moreover, osteoblast-derived EVs have been shown to function as nucleation sites for mineralization. By providing phosphate through ALP activity and initiating calcium phosphate crystal formation, these EVs drive the mineralization of the bone matrix, supported by in vitro and in vivo models [100,101]. Collectively, these findings suggest that osteoblasts, through EV-mediated mechanisms, remodel the bone matrix to enrich growth factors, thereby creating a microenvironment favorable for prostate cancer progression (Figure 2).

In PCa bone metastases, osteoblasts are not merely bone-forming cells but key regulators of the metastatic microenvironment. Osteoblast-derived EVs contribute in three major ways: (i) transmitting signals to tumor cells that can either suppress or promote malignancy, (ii) modulating osteoclast differentiation through both inhibitory and stimulatory pathways, and (iii) directly reshaping the bone microenvironment by driving matrix remodeling and initiating mineralization. Together, these functions support the osteoblastic phenotype characteristic of prostate cancer bone metastases and highlight EV-mediated pathways as potential therapeutic targets.

## 5. The Direction of EV Research for Developing Novel Therapeutic Approach Toward Bone Metastasis

Focusing EV-mediated crosstalk between tumor cells and the bone microenvironment during the progression of PCa bone metastasis, it is becoming increasingly evident that deleterious EVs involved in metastatic progression represent promising therapeutic targets. One strategy involves suppressing the production of tumor-derived EVs, particularly those that drive bone metastasis. EV biogenesis generally follows two main pathways: the endosomal sorting complex required for transport (ESCRT)-dependent pathway and the ESCRT-independent pathway [102]. Among the various EV subtypes, exosomes which are formed via inward budding within early endosomes and subsequently released through fusion of multivesicular bodies (MVBs) with the plasma membrane have been the most extensively studied [103,104]. Representative EV marker proteins such as ALG-2-interacting protein X (ALIX) and Tumor susceptibility gene 101 protein (TSG101) are associated with the ESCRT-dependent mechanism [105]. On the other hand, the ESCRT-independent pathway involves ceramide, a lipid implicated in MVB biogenesis, and its synthesis is regulated by neutral sphingomyelinase 2 (nSMase2) [106].

Cancer therapies targeting general EV production pathways are actively being investigated. Kosaka et al. reported that knockdown of nSMase2 reduced EV production and suppressed miR-210 expression, thereby inhibiting angiogenesis and metastasis in a xenograft mouse model [54]. Similarly, attempts have been made to inhibit EV production in cancer cells using GW4869, a chemical inhibitor of nSMase2; however, this approach also suppresses EV production in normal cells, presenting a major limitation. Therefore, identifying cancer cell–specific EV secretion mechanisms is critical. Our group previously demonstrated that miR-26a regulates EV secretion from the bone-metastatic prostate cancer cell line PC3M by targeting SHC adaptor protein 4 (SHC4), Prefoldin subunit 4 (PFDN4), and Cysteine and histidine-rich domain (CHORDC)1, and that this suppression attenuates tumor growth, supported by in vitro and in vivo models [107]. Additionally, Yamamoto et al. identified the miR-891b/Phosphoserine Aminotransferase 1 (PSAT1) axis, involved in the serine–ceramide synthesis pathway, as a regulator that enhances EV secretion from cancer cells, supported by in vitro studies [108]. Notably, PSAT1-mediated cancer-derived EVs promote osteoclast differentiation and accelerate bone metastasis, highlighting this pathway as a potential therapeutic target for inhibiting metastatic progression, supported by in vitro and in vivo models [108]. These factors are particularly attractive as therapeutic targets, as they may inhibit bone metastasis without adversely affecting normal cells.

Another promising approach involves the removal of circulating deleterious EVs [109]. Selective elimination of Human Epidermal growth factor Receptor 2 (HER2)-positive EVs, which are known to interfere with drug efficacy and promote tumor progression, has shown therapeutic benefit in HER2-positive breast cancer, supported by in vitro and in vivo models [110]. In a metastatic human xenograft breast cancer mouse model, Nishida et al. demonstrated that co-administration of therapeutic antibodies with human-specific anti-CD9 and anti-CD63 antibodies led to efficient removal of tumor-derived EVs by macrophages, resulting in a dramatic reduction in tumor burden [111]. These studies introduce novel EV-targeted therapeutic concepts. Given the ongoing extensive research on prostate cancer–associated membrane proteins such as Prostate Specific Membrane Antigen (PSMA), these strategies have strong potential to be translated into therapies for inhibiting prostate cancer metastasis, supported by in vitro, in vivo, and clinical observations [112,113,114].

As a conceptually distinct therapeutic strategy, the use of EVs as vehicles for drug delivery has also attracted considerable attention [115]. While molecular targeted therapies including antibody and nucleic acid-based drugs are being actively developed worldwide, their therapeutic efficacy is often limited by poor delivery efficiency to target lesions. Therefore, the development of innovative drug delivery systems (DDS) capable of delivering sufficient drug concentrations to the local lesion is essential for effective clinical application. EVs represent ideal DDS candidates due to their small size and prolonged circulation time.

Currently, docetaxel is the most widely used chemotherapeutic agent with proven clinical efficacy against PCa bone metastases. However, systemic administration of docetaxel is associated with significant toxicity to normal cells. Tian et al. successfully encapsulated the chemotherapeutic drug doxorubicin (DXR) into small EVs derived from murine immature dendritic cells (imDCs) using electroporation. These imDCs were engineered to overexpress the EV membrane protein Lamp2b fused with an αv integrin-specific peptide to enhance targeting of αv integrin-expressing cells, such as MDA-MB-231 breast cancer cells. In a breast cancer mouse model, these engineered sEVs exhibited potent tumor-suppressive effects with reduced toxicity compared to free DXR administration. Although this study did not focus on bone metastases, it demonstrates the therapeutic potential of EV-based DDS in treating patients with multiple metastatic lesions [116].

Cappariello et al. further demonstrated that osteoblast-derived EVs can efficiently internalize and deliver anti-osteoclastic drugs such as zoledronate, thereby suppressing osteoclast activity both in vitro and in vivo [99]. This suggests that osteoblast-derived EVs could be exploited to inhibit tumor-induced osteolysis, providing a potential therapeutic strategy for PCa bone metastases. Because EVs from resident bone cells are likely to act specifically within the bone microenvironment, they are of particular interest as DDS carriers specialized for bone metastasis therapy. However, challenges such as safety and scalability remain for clinical application, underscoring the need for further research and development.

## 6. Conclusions

EVs are now recognized as key mediators of tumor–bone communication in PCa. Unlike other solid tumors, PCa develops predominantly osteoblastic bone metastases, which are suggested to function as a hub for systemic disease progression. Tumor-, osteoclast-, and osteoblast-derived EVs cooperatively regulate osteolytic and osteoblastic processes, remodel the bone matrix, and influence tumor cell behavior, thereby shaping the unique pathophysiology of PCa bone metastases. Importantly, EVs exhibit dual, microenvironment- and cargo-dependent functions, serving either as promoters or suppressors of malignancy. These insights highlight EVs not only as mechanistic drivers but also as promising therapeutic targets, with potential strategies such as inhibiting tumor-derived EV production, removing harmful circulating EVs, and utilizing EVs as drug delivery systems.

## Figures and Tables

**Figure 1 ijms-26-12160-f001:**
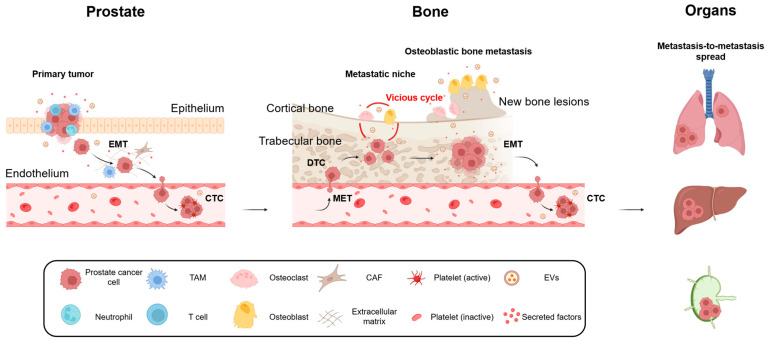
The stepwise process of prostate cancer bone metastasis. Schematic illustration showing the multistep process of prostate cancer dissemination and colonization in the bone microenvironment. Tumor cells undergo epithelial–mesenchymal transition (EMT) and intravasate into the circulation as circulating tumor cells (CTCs). CTCs home to the bone marrow and settle as disseminated tumor cells (DTCs), where they interact with stromal and bone-resident cells to form a metastatic niche. Within the bone microenvironment, cancer cells establish a vicious cycle by stimulating osteoclast-mediated bone resorption and releasing growth factors that in turn promote tumor proliferation. Subsequently, osteoblast activation leads to the development of osteoblastic bone metastases, a hallmark of prostate cancer. Metastasis-to-metastasis spread from established bone lesions to distant organs and lymph nodes is also illustrated. EMT, epithelial–mesenchymal transition; MET, mesenchymal–epithelial transition; CTC, circulating tumor cell; DTC, disseminated tumor cell.

**Figure 2 ijms-26-12160-f002:**
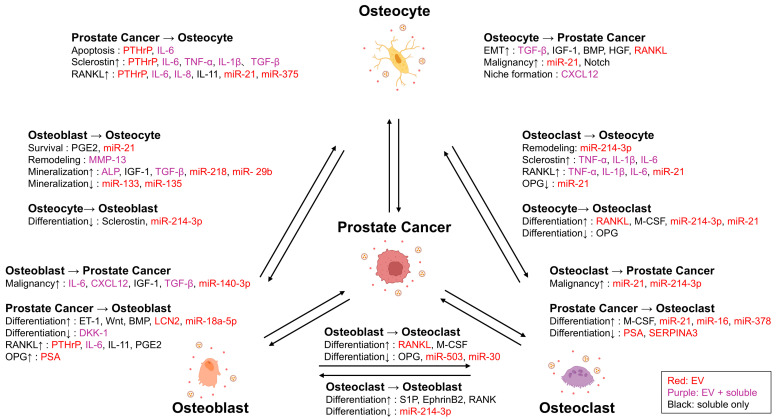
Extracellular vesicle and cytokine-mediated crosstalk in prostate cancer bone metastasis. Schematic representation of the complex crosstalk between PCa cells and bone-resident cells, including osteoblasts, osteoclasts, and osteocytes. Red text indicates molecules that are specifically enriched in EVs. Purple text indicates molecules that can be found in both EVs and soluble form. Black text indicates molecules that are primarily released as soluble factors with no strong evidence of EV encapsulation.

## Data Availability

No new data were created or analyzed in this study. Data sharing is not applicable to this article.

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
