# Peer review of "Significance of EVs in Prostate Cancer Bone Metastases"

_ijms, 2025, doi:10.3390/ijms262412160_

Round 1

Reviewer 1 Report

Comments and Suggestions for Authors

In the manuscript entitled “Significance of EVs in Prostate Cancer Bone Metastases” the authors present a comprehensive review highlighting the pivotal role of extracellular vesicles (EVs) in the bidirectional communication that regulates osteoclastogenesis, osteoblast activation, osteocyte function, and prostate cancer progression. Based on an extensive evaluation of the current literature, the authors conclude that EVs act not only as mechanistic drivers of prostate cancer bone metastasis but also as promising therapeutic targets. The subject of this article is highly interesting and relevant for a better understanding of the complex and dynamic processes underlying prostate cancer development, as well as for the advancement of potential drug discovery, drug delivery, and therapeutic strategies.

Some minor revisions should be addressed before acceptance for publication in the journal International Journal of Molecular Sciences.

  1. Terms for all abbreviations used in the manuscript should be provided (i.e. IL-6, IGF-1 etc…). Also in Figure 1 MET abbreviation explanation is missing.
  2. The authors could consider introducing couple of sentences about prostasomes (EVs originated from prostate) which could be beneficial in introduction section.
  3. Page 6 – “Tumor-derived extracellular vesicles (EVs) have attracted attention as critical modulators of osteoblastic lesion development – delete extracellular vesicles, “Tumor-derived EVs” is fine
  4. Page 9 – Add reference [100] after sentence: Additionally, Yamamoto et al. identified the miR-891b/PSAT1 axis, involved in the serine–ceramide synthesis pathway, as a regulator that enhances EV secretion from cancer cells.

Reviewer 2 Report

Comments and Suggestions for Authors

In the present review, the authors attempt to summarize the data on the role of extracellular vesicles in prostate cancer. Although the intention is to focus in extracellular vesicles, the manuscript rather gives an overall review of prostate cancer metastasis, with a relatively limited focus on the vesicles. 

If the intention is indeed to highlight the role of EVs, my opinion is that the manuscript is rewritten, with the general parts of prostate cancer metastasis and the role of osteoclasts and osteoblasts shortened and fused to one section. A separate section should elaborate on EVs in cancer. Then, a third section on EVs in prostate cancer.

In every section, it should be made clear for the reader which studies report on in vitro data, which on animal models and which on clinical. 

Other points: 

Figure 2 is very difficult to follow. A separate Figure on EVs formation in cancer would add to the paper. 

Comments on the Quality of English Language

Only occasional,  minor errors detected

Round 2

Reviewer 2 Report

Comments and Suggestions for Authors

In the revised version the authors introduced information regarding the original studies which makes interpretation of the data much easier. The article has thus improved. Figure 2 is still without any value for the reader, and the argument that there are other relevant in the literature is weak, there should be an effort to have a figure that is applicable to the present manuscript. 

Comments on the Quality of English Language

Only occasional,  minor errors detected
